# Differentially Expressed Genes in Cardiomyocytes of the First Camelized Mouse Model, Nrap^c.255ins78^ Mouse

**DOI:** 10.3390/genes16020142

**Published:** 2025-01-24

**Authors:** Sung-Yeon Lee, Byeonghwi Lim, Bo-Young Lee, Goo Jang, Jung-Seok Choi, Xiang-Shun Cui, Kwan-Suk Kim

**Affiliations:** 1Department of Animal Sciences, Chungbuk National University, Cheongju 28644, Republic of Korea; sypolaris@chungbuk.ac.kr (S.-Y.L.); jchoi@chungbuk.ac.kr (J.-S.C.); xscui@chungbuk.ac.kr (X.-S.C.); 2Laboratory of Theriogenology, Department of Veterinary Clinical Science, College of Veterinary Medicine, Seoul National University, Seoul 08826, Republic of Korea; snujang@snu.ac.kr; 3Department of Animal Science and Technology, Chung-Ang University, Anseong 17546, Republic of Korea; hwi1208@cau.ac.kr; 4Department of Biological Science, University of New Hampshire, Durham, NH 03824, USA; bo-young.lee@unh.edu

**Keywords:** adaptation, mouse model, heart, transcriptome *Nrap*

## Abstract

**Background/Objectives:** The first camelized mouse model (*Nrap^c.255ins78^*) was developed to investigate the mechanisms underlying camels’ adaptation to extreme environments. Previous studies demonstrated that these mice exhibit a cold-resistant phenotype, characterized by increased expression of inflammatory cytokine-related genes in the heart under cold stress. Nebulin-related anchoring protein (*NRAP*) plays a critical role in organizing myofibrils during cardiomyocyte development. This study builds on prior research by analyzing the heart transcriptomes of *Nrap^c.255ins78^* mice under non-stress conditions to explore the origins of inflammatory cytokine responses during cold exposure. **Methods:** RNA sequencing was performed on the hearts of 12-week-old male and female *Nrap^c.255ins78^* and wild-type control mice. **Results:** Differential expression analysis identified 25 genes, including 12 associated with cell cycle and division, all consistently downregulated in *Nrap^c.255ins78^*. Notably, the calcium and integrin-binding protein gene (*Cib3*) was significantly upregulated (FDR < 0.05; *p* < 0.001). **Conclusions:** These differentially expressed genes suggest altered calcium dynamics in cardiomyocytes and mechanisms for maintaining homeostasis, supporting the hypothesis that inflammatory cytokines during cold exposure may represent an adaptive response. These findings provide valuable insights into the genetic mechanisms of temperature adaptation in camels and highlight potential pathways for enhancing stress resistance in other mammals.

## 1. Introduction

Camels are unique mammals that have adapted remarkably to extreme environmental conditions, thriving in harsh habitats like deserts [1]. Camelid genomes have been studied to uncover the genetic mechanisms driving their evolution and adaptation to harsh desert environments [2,3]. However, our understanding of the genetic variations that drive environmental adaptation in camels remains incomplete.

Our previous research identified that exon 4 of the *NRAP* gene in camels plays a significant role in cold resistance. Specifically, the *Nrap^c.255ins78^* variant in mice exhibits a unique expression in the heart, where it enhances inflammatory cytokine production under cold stress, thereby contributing to temperature resilience [4]. *NRAP* plays a vital role in muscle structure and function, with a mutation at amino acid residue 100 potentially contributing to survival in extreme northern environments [5], and cytokine expression is known to change in mammals exposed to cold [6,7]. For instance, cold exposure has been reported to increase interleukin (IL)-1β and IL-6 levels in cold-resistant humans [8], while in mice, cold stress elevates tumor necrosis factor (TNF)-α and IL-6 levels [9].

These findings suggest a greater upregulation of inflammatory cytokines in *Nrap^c.255ins78^* mice compared to wild-type mice. To investigate these cytokine mechanisms, this study analyzes changes in the heart transcriptome of *Nrap^c.255ins78^* mice under non-stress conditions using RNA sequencing (RNA-seq). RNA-seq is a widely used tool for analyzing transcriptomic changes in organs influenced by internal and external environments [10,11,12,13]. This approach, comparing the differentially expressed gene set under normal conditions, may provide insights into the cytokine expression pathways observed in the heart under cold stress. Ultimately, this study could further contribute to our understanding of temperature adaptation mechanisms in camels and environmental stress adaptation in various mammals.

## 2. Materials and Methods

### 2.1. Animals

The *Nrap^c.255ins78^* mice were generated by the GEM center in Macrogen Inc. (Seoul, Republic of Korea), according to a previous study [4], while the C57BL/6N mice were purchased from Orientbio Inc. (Sungnam, Republic of Korea). Both the *Nrap^c.255ins78^* and the wild-type mice, which were used as controls, have a C57BL/6N background and were housed at the GEM center of Macrogen Inc. in a specific pathogen-free environment. The temperature and humidity of the breeding environment were maintained at 22 ± 1 °C and 50%, respectively, with a 12 h light/dark cycle. All feed, individually ventilated cages, and air conditioners were sterilized, with the air being filtered through strainers.

### 2.2. RNA-Seq

To investigate gender-based differences in gene expression, twelve-week-old *Nrap^c.255ins78^* homozygous mice (2 males and 2 females) were used, along with twelve-week-old wild-type mice (2 males and 2 females) as controls. The excised hearts were photographed to confirm phenotypic characteristics, and total RNA was extracted using Trizol reagent (Sigma-Aldrich, St. Louis and Burlington, MA, USA) following the manufacturer’s instructions. Detailed RNA sequencing procedures were reported previously [4].

### 2.3. Differentially Expressed Gene (DEG) Analyses

The quality of raw read data for each sample was assessed using FastQC software v0.11.7. Adaptor trimming was conducted with Trimmomatic v0.38, based on quality results. The trimmed reads were then aligned to the reference genome (GRCm39) from the Ensembl genome browser using HISAT2 v2.1.0. Raw counts for each library were calculated based on the exons in Mus musculus GTF v110 (Ensembl) using the featureCounts function of the Subread package v1.6.3. DEG analysis was conducted using edgeR v3.26.5, with raw counts normalized via the TMM (Trimmed Mean of M-values) method. DEGs were identified in the hearts of *Nrap^c.255ins78^* mice compared to wild-type mice, using a false discovery rate (FDR) of < 0.05 and an absolute log_2_ fold-change (FC) threshold of ≥ 1. A multidimensional scaling (MDS) plot analysis was also performed to illustrate the sample clustering.

### 2.4. Gene Ontology (GO) Functional Enrichment Analysis

The identified DEGs were annotated with GO terms using DAVID (Database for Annotation, Visualization, and Integrated Discovery) v2024q1 [14]. GO annotations were conducted across the categories of Biological Processes (BP), Cellular Components (CC), and Molecular Functions (MF), using thresholds of *p*-value < 0.05 and counts ≥ 2. Enriched GO terms were grouped with similar terms and visualized in a bubble plot, displaying −log_10_
*p*-value and fold enrichment.

### 2.5. Interaction Network Construction

A network based on the DEGs was constructed using the Search Tool for the Retrieval of Interacting Genes (STRING) in Cytoscape v3.10.2. The interaction score was set to 0.4, representing a medium confidence level.

### 2.6. RT-qPCR Validation

Reverse transcription–quantitative polymerase chain reaction (RT-qPCR) was performed using the SYBR Green PCR mix (Qiagen, Hilden, Germany) to validate the RNA-seq results. *Actb* (β-actin) was used as a control for normalization. The relative quantification of mRNA expression was calculated using the 2^−ΔΔCt^ method, and results were presented as the average relative fold change. The primer sequences and amplification temperatures are provided in Appendix A.

### 2.7. Statistics

RNA-seq data analysis and visualization were conducted using R statistical software v4.3.3. The qPCR experiments were independently repeated three times, and the values were presented as mean ± SD. Statistical significance was evaluated using a paired *t*-test, with a *p*-value of less than 0.05 considered statistically significant.

## 3. Results

### 3.1. Data Processing and Transcriptomes

The average overall mapping rate was 99.15%, and the average unique mapping rate was 72.60% (Appendix A). Multiclustering was observed in the MDS analysis, with the transcriptomes of each sample showing differentiation by sex and genotype (Figure 1A). DEG analysis was conducted by comparing expression levels, which were visualized in a volcano plot (FDR < 0.05, absolute log_2_FC ≥ 1) (Figure 1B). In total, 25 DEGs were identified in the heart tissue, with 7 upregulated and 18 downregulated (Appendix A).

### 3.2. Functional Annotations

Functional enrichment analysis was conducted based on GO terms in heart tissue and visualized with a bubble plot (Figure 1C). The most significantly enriched BP included cell division (GO:0051301; *p*-value = 8.78 × 10^−10^; FDR = 1.21 × 10^−7^) and cell cycle (GO:0007049; *p*-value = 1.30 × 10^−9^; FDR = 1.21 × 10^−7^). A comprehensive list and detailed values for each GO term are provided in Appendix A.

### 3.3. Expression Pattern and Validation

We visualized the expression patterns of the 25 DEGs across all samples in a heatmap (Figure 2A) and performed a co-expression network analysis to examine interactions among genes identified in enriched GO terms (Figure 2B). The network analysis revealed interactions based on the GO terms “cell division” and “cell cycle”. Within this network, the *Kif11* gene displayed a text-mining association with *Cib3*, which was identified as an upregulated gene.

To confirm the reliability of the RNA-seq data, we selected three upregulated genes (*Cib3*, *Aldob*, *Rtn4r*), four downregulated genes (*Ccnb1*, *Kif11*, *Aspm*, *Ncapg*), and two non-significant genes (*Nrap*, *Spkh2*) for validation. The expression levels of all genes closely matched the transcriptome analysis results (Figure 2C).

## 4. Discussion

In mammals, the heart is a critical organ for regulating body temperature and circulating blood to accommodate sudden physiological changes [15,16,17]. In this study, we grouped the heart transcriptomes of *Nrap^c.255ins78^* and wild-type mice by genotype and sex.

MDS visualization revealed transcriptomic differences across the groups (Figure 1A), and the distinct expression patterns grouped by *Nrap* genotype and biological sex suggest that *Nrap* expression may have different effects depending on sex. Understanding how genetic variations influence sex-specific cardiac physiology will be highly significant [18], and gene expression differences based on sex remain an important area for further investigation in future studies. In this study, we focused on examining expression differences influenced by the *Nrap* genotype.

Functional annotation of the 25 DEGs revealed significant associations with the GO terms “cell cycle” and “cell division” (Figure 1C). Notably, all 12 genes (*Kif11*, *Anln*, *Ccnb1*, *Kntc1*, *Kif23*, *Ncapg*, *Aspm*, *Mki67*, *Pimreg*, *Knl1*, *Top2a*, *Nr4a3*) associated with cell cycle and division terms were consistently downregulated in *Nrap^c.255ins78^* mice (Figure 2A,B). *Nr4a3* enhances glycolytic activity by directly binding to the promoter regions of two glycolysis-related genes, *Aldoa* and *Pfkl*, thereby driving their transcriptional initiation [19]. Given the heart’s continuous workload and its high demand for energy metabolism [17], the reduced expression of *Nr4a3* in *Nrap^c.255ins78^* mice may contribute to metabolic differences compared to wild-type mice.

The DEG network highlights interactions among 12 genes involved in cell cycle and division, with *Cib3* (calcium and integrin-binding protein 3) linked to *Kif11* (Figure 2B). *Cib3* was the most significantly upregulated gene in *Nrap^c.255ins78^* mice (Figure 2C, Appendix A) and is part of the CIB protein family, which includes *Cib1*, *Cib2*, and *Cib4* [20]. CIB proteins are small EF-hand calcium-binding proteins that interact with a wide range of biological targets [21]. Notably, both *Cib2* and *Cib3* show expression in certain overlapping tissues, including the heart [22]. CIB proteins in various tissues across organisms suggest involvement in diverse biochemical processes, initially linked to the interaction with not only integrin but also many other proteins in intracellular and extracellular signaling [23,24,25,26,27]. CIB3 protein can functionally substitute for CIB2 [23]. While the functions of other CIB proteins are better understood, *Cib3* remains relatively unexplored, making its high expression levels in *Nrap^c.255ins78^* mice particularly intriguing—especially since no differences were observed in the expression levels of *Cib1* and *Cib2*. The amino acid homology between camel and mouse CIB3 was found to be exceptionally high at 92%, with the coding sequence also being identical at 564 bp (Appendix A). This suggests the possibility that the same mechanism observed in *Nrap^c.255ins78^* mice could also occur in camels.

Inflammatory cytokines have been observed in mammals exposed to cold stress [6,7,8,9] and are known to regulate calcium influx by modulating calcium levels within cellular organelles and the nucleus [28,29]. Calcium signaling begins with the influx of calcium from the extracellular space or its release from the endoplasmic reticulum or mitochondria [30]. Calcium influx into the mitochondria regulates the citric acid cycle to promote ATP production [31]. Maintaining appropriate calcium levels in the heart is crucial for preserving cytoplasmic calcium homeostasis through ATPase pumps, as excessive calcium in cardiomyocytes can increase the heart rate, leading to cell necrosis and apoptosis [32,33]. Although our analysis alone cannot establish a causal relationship between cell proliferation and *Cib3* upregulation, the observed inhibition of cell cycle and division genes in *Nrap^c.255ins78^* mouse hearts suggests altered calcium intracellular signaling pathways [30,34].

The transcriptome analysis did not identify a differential expression of the *Nrap* gene, but the qPCR results indicated a slight increase in its expression (Figure 2C). The NRAP is a Z-disk protein with nebulin-like super-repetitive sequences, and it plays a crucial role in myofibril formation [35,36]. Nebulin can bind to calmodulin (CaM) and regulate calcium release in the skeletal muscle [37,38]. The predictions of the domains and protein structure of *Nrap^c.255ins78^* suggest that there are structural and functional alterations in the protein [4]. Although further in-depth analysis of the protein structure is necessary, this change in the domain structure of the *Nrap^c.255ins78^* protein may lead to altered interactions with proteins like CaM, potentially affecting calcium levels in cardiomyocytes [39].

Transgenic mice with a cardiac-specific overexpression of the human DSC2 developed severe cardiomyopathy shortly after birth, with significantly reduced fractional shortening and ejection fractions [40]. Despite the downregulation of genes involved in cell cycle and division, the hearts of *Nrap^c.255ins78^* mice did not significantly differ in size from those of wild-type mice (Appendix A). Our findings show that downregulated genes such as *Angptl4*, *Ccnb1*, *Mki67*, and *Top2a* are known to be influenced by YAP/TAZ signaling, and they function as transcriptional co-activators, primarily mediating the Hippo signaling pathway but also involved in other signaling pathways that control tissue growth and organ size [41,42]. While the precise mechanisms of YAP/TAZ remain unclear, compensatory effects on actin cytoskeleton stability may have helped to maintain organ size under these conditions [43]. Based on these findings, future studies should include histological staining, heart rate measurements, and blood composition analyses, such as calcium concentration assessments, to better understand the physiological effects of the molecular mechanisms in *Nrap^c.255ins78^* mice when trying to emulate camels’ cold adaptation.

## 5. Conclusions

The heart transcriptomes of *Nrap^c.255ins78^* mice under non-stress conditions revealed consistent downregulation of cell proliferation-related genes alongside a significant upregulation of the *Cib3* gene. These findings suggest a homeostatic response within cardiomyocytes to altered calcium dynamics, ultimately indicating heightened cytokine sensitivity compared to the wild-type group. This study provides foundational insights into temperature adaptation mechanisms, broadening the understanding of the *NRAP* gene’s exon 4 function beyond camels.

## Figures and Tables

**Figure 1 genes-16-00142-f001:**
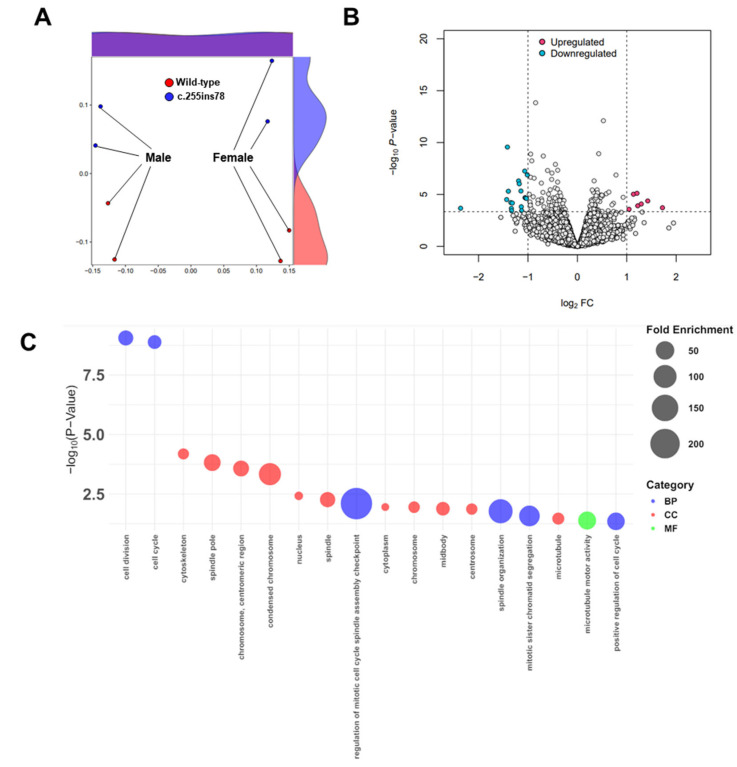
Transcriptome in the heart of *Nrap*^c.255ins78^ and wild-type mice. (**A**) Group clustering and multidimensional scaling (MDS). Grouping transcript frequencies between *Nrap^c.255ins78^* and wild- type mice revealed distinct differences. (**B**) Differentially expressed genes (DEGs) identified using a volcano plot are displayed, with the *x*-axis representing the log_2_ fold change and the *y*-axis representing the -log_10_ *p*-value. (**C**) The Gene Ontology (GO) bubble plot was created based on the -log *p*-values and fold enrichment related to the biological processes (BP), cellular components (CC), and molecular functions (MF) terms.

**Figure 2 genes-16-00142-f002:**
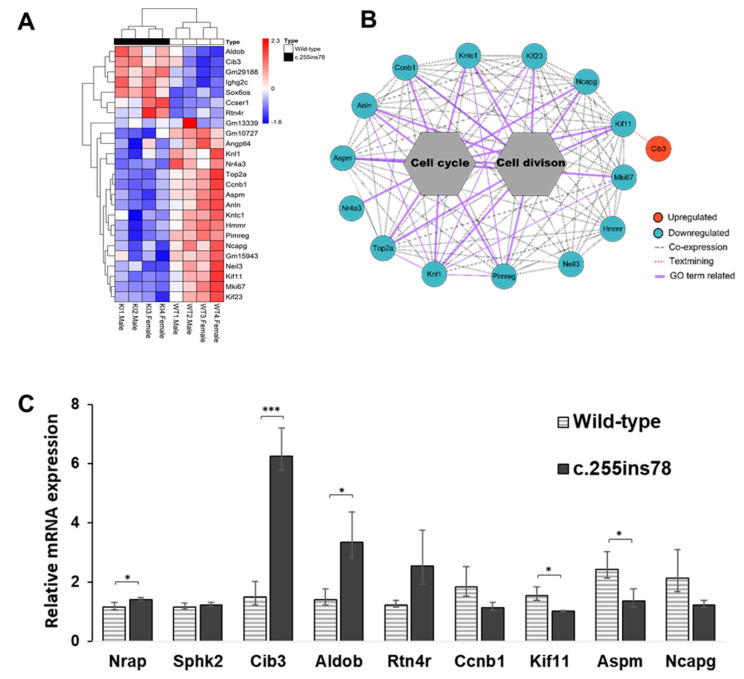
Expression pattern of *Nrap^c.255ins78^* heart. (**A**) Heatmap for expression between all differentially expressed genes (DEGs). (**B**) The interactions using DEGs indicate the associations between the co-expression of each gene. (**C**) Verification of the RNA-seq by RT-qPCR; * and *** indicate *p*-value < 0.05 and *p*-value < 0.001, respectively.

## Data Availability

We have submitted the RNA-seq data to the National Center of Biotechnology Information’s Sequence Read Archive under accession number PRJNA1175414.

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
