# Peer review of "Differentially Expressed Genes in Cardiomyocytes of the First Camelized Mouse Model, Nrap^c.255ins78^ Mouse"

_genes, 2025, doi:10.3390/genes16020142_

Round 1
Reviewer 1 Report
Comments and Suggestions for Authors
In the original article 'Differentially Expressed Genes in Cardiomyocytes of the First Camelized Mouse Model, Nrapc.255ins78 Mouse' the authors investigate using RNA-Seq of the heart a differentially expressed gene cluster in a camelized mouse model (Nrap c.255ins78). The topic of this article is interesting. However, I suggest some changes and extensions:
1.) Please add two sentences about the function of the Nrap gene within the abstract.
2.) Could you add also a species alignment of Nrap to show differences between mouse, camels and humans?
3.) Line 105: Please use the official gene name for ß-actin.
4.) I suggest to present all data as mean +/- SD instead of SEM, since the reader is interested in the variation of the data. Larger error bars are not a problem.
5.) Is it possible to investigate by Western blots, if the transcriptomic RNA changes cause also a regulation of the protein? I know that this is a lot of work, but from my perspective this would improve the quality of your manuscript!
6.) Inflammatory cytokine-related changes are also present in mouse models for genetic cardiomyopathies (see ‘Transgenic mice overexpressing desmocollin-2 (DSC2) develop cardiomyopathy associated with myocardial inflammation and fibrotic remodeling’, 2017 PLOSone). Could you discuss this point?
In summary, I suggest a major revision.
Author Response
1) Please add two sentences about the function of the Nrap gene within the abstract.
Thank you for your comment. We have added a brief description of the function of NRAP. Please refer to lines 21-22 of the revised manuscript for the updated information.
2) Could you add a species alignment of Nrap to show differences between mouse, camels, and humans?
In our previous study, we analyzed the NRAP amino acid sequences across 155 mammalian species, including mice, camels, and humans, and reported that only camels possess Exon 4 (PMID: 39478283). We have also attached the amino acid alignment results for mice, camels, and humans. Please refer to the supplementary figure (Align of three species_genes-3431183) in the attached PNG file.
3) Line 105: Please use the official gene name for ß-actin.
Thank you for your valuable feedback. We have updated the manuscript by changing "ß-actin" to its official name, "Actb," in Line 111 and Table S1.
4) I suggest presenting all data as mean ± SD instead of SEM, since the reader is interested in the variation of the data. Larger error bars are not a problem.
Thank you for your suggestion. We have revised the manuscript and applied SD to the qPCR data. Please refer to lines 119-120 and Figure 2C of the revised manuscript for the changes.
5) Is it possible to investigate by Western blot whether the transcriptomic RNA changes also regulate the protein? I know this is a lot of work, but from my perspective, this would improve the quality of your manuscript!
We appreciate the importance of Western blot analysis to verify the absence of truncated proteins or assess the level of expression. However, in our previous study, we confirmed normal expression at the RNA level (PMID: 39478283). The inserted sequence consists of 26 in-frame residues, and based on its proper transcription, we assume that the translation of NRAPc.255ins78 proceeds normally. We will certainly consider your suggestion to perform a Western blot in future studies to determine whether this protein is expressed at higher levels. We agree that functional studies at the protein level are essential, and we greatly appreciate your valuable input.
6) Inflammatory cytokine-related changes are also present in mouse models for genetic cardiomyopathies (see ‘Transgenic mice overexpressing desmocollin-2 (DSC2) develop cardiomyopathy associated with myocardial inflammation and fibrotic remodeling’, 2017 PLOS One). Could you discuss this point?
Thank you for your insightful comment. We have included and discussed the study you referenced in lines 221-223 of the revised manuscript.
We sincerely appreciate your thoughtful comments and kindly ask for your consideration of the points discussed above.
Reviewer 2 Report
Comments and Suggestions for Authors
Summary
In order to elucidate the genetic pathways underpinning environmental stress adaptation, this paper examines the heart transcriptomes of the camelized mouse model (Nrapc.255ins78). With an emphasis on calcium signaling and inflammatory cytokine responses, the scientists use RNA sequencing to find differentially expressed genes (DEGs) between the Nrapc.255ins78 and wild-type mice in non-stressful settings. The study lays the groundwork for future investigations into stress adaption mechanisms and offers insightful information about possible pathways influencing temperature resilience.
Strengths
The camelized mouse model is a creative way to study environmental adaptability.
Strong bioinformatics approaches are demonstrated in the research through the use of RNA sequencing, network analysis, and GO enrichment analysis.
The goal of understanding temperature adaptation aligns with the focus on cytokine responses and calcium signaling.
RT-qPCR validation of RNA-seq results increases the results' dependability.
Recommendations
Figure 1resolution is low, consider providing higher-resolution images with clearer annotations.
How do the observed changes in Cib3 and related genes mechanistically impact cardiomyocyte function?
Could calcium dysregulation explain the increased inflammatory cytokine production observed under cold stress in earlier studies?
No functional assays were performed to validate the proposed roles of DEGs in calcium dynamics or cell cycle regulation. Could the authors explore in vitro cardiomyocyte models or knockout experiments to substantiate these findings?
The manuscript does not elaborate on its biological implications. Are there specific DEGs that are predominantly influenced by sex? How might these differences impact the interpretation of the data?
While the study links Nrapc.255ins78 to camel adaptation mechanisms, it does not directly demonstrate how these findings translate to camels. Could the authors include a discussion of the homologous genes or pathways in camels to strengthen the connection?
The small sample size (n=4 per group) may reduce statistical power. How do the authors account for this limitation?
The study focuses solely on the heart. Would transcriptomic analysis of other organs, such as lungs or kidneys, provide a more comprehensive view of temperature adaptation?
Conclusion
Using a novel camelized mouse model, this study makes a significant advance to our understanding of the genetic underpinnings of temperature adaptation. But resolving problems with figure quality, mechanistic insights, functional validation, and constraints will greatly improve the study's effect and clarity. The paper will become stronger and more publishable with these changes.
Author Response
- Figure 1 resolution is low, consider providing higher-resolution images with clearer annotations.
Thank you for your valuable comment. We have improved the resolution of Figure 1 and clarified the annotations. Please refer to Figure 1 for your review. - How do the observed changes in Cib3 and related genes mechanistically impact cardiomyocyte function?
Thank you for your constructive question. We have revised the manuscript to discuss the functions of Cib3 in lines 191-196. - Could calcium dysregulation explain the increased inflammatory cytokine production observed under cold stress in earlier studies?
The calcium-dependent upregulation of Cib3 and the downregulation of cell cycle-regulating genes reflect a response to disrupted calcium influx in cardiomyocytes, which may have made them more sensitive to inflammatory cytokines under cold stress conditions. Please refer to lines 200-211 of the revised manuscript for further details. - No functional assays were performed to validate the proposed roles of DEGs in calcium dynamics or cell cycle regulation. Could the authors explore in vitro cardiomyocyte models or knockout experiments to substantiate these findings?
We agree with your thoughtful feedback. Additional studies are planned, as discussed in lines 233-236. We aim to establish an in vitro cardiomyocyte model using primary cultures of mouse cardiomyocytes to further elucidate the underlying mechanisms. We sincerely appreciate your insightful comments. - The manuscript does not elaborate on its biological implications. Are there specific DEGs that are predominantly influenced by sex? How might these differences impact the interpretation of the data?
Our initial research focused on minimizing sex-based variables to conduct an integrated analysis, so DEG selection based on sex differences was not initially considered. However, through multidimensional scaling (MDS) analysis based on the integrated transcriptome, we observed differences not only by genotype but also by the biological sex of the mice within the groups.
Thank you very much for your insightful feedback. Please refer to the rephrased sentences in lines 167–173 of the revised manuscript for further discussion. - While the study links Nrapc.255ins78 to camel adaptation mechanisms, it does not directly demonstrate how these findings translate to camels. Could the authors include a discussion of the homologous genes or pathways in camels to strengthen the connection?
We greatly appreciate your constructive and meaningful feedback. This study serves as a foundation for future detailed investigations. We acknowledge the limitation that our current results do not directly demonstrate how these findings could be applied to camels.
Currently, research on the specific molecular mechanisms underlying environmental adaptation, particularly cold adaptation, in camels is limited. However, based on our foundational findings, our ultimate goal is to better understand the detailed mechanisms of both cold and heat adaptation in camels and to gain insights applicable to livestock and other animals. Please refer to the rephrased sentences in lines 196-199 and 233–236 of the revised manuscript for further details. - The small sample size (n=4 per group) may reduce statistical power. How do the authors account for this limitation?
We fully acknowledge your concerns and have taken several measures to minimize variability. Specifically, we strictly standardized the experimental conditions, including the rearing environment and age, and employed stringent cut-off criteria (FDR < 0.05; P < 0.001) for DEG selection. As a result, we identified only 25 DEGs, reflecting a conservative approach.
Additionally, to ensure the reliability of our findings, we conducted RT-qPCR validation with three technical replicates per sample, which strengthened the credibility of our results.
In future studies, we aim to increase the sample size per group to further enhance the robustness and reliability of our findings. We kindly ask you to consider that this study serves as an essential foundation for further research in this field. - The study focuses solely on the heart. Would transcriptomic analysis of other organs, such as lungs or kidneys, provide a more comprehensive view of temperature adaptation?
Thank you for your valuable feedback. We are also considering transcriptomic analyses of the lungs and kidneys as part of our future research. We believe that changes in the heart, as the central organ in the mammalian circulatory system, will inevitably be reflected in the lungs or kidneys. We trust that this study will provide foundational data to interpret such changes.
Round 2
Reviewer 1 Report
Comments and Suggestions for Authors
The authors have improved their manuscript according to my previous comments. Therefore I suggest to accept this manuscript for publication.